# EXPLICITLY MINIMIZING THE BLUR ERROR OF VARIATIONAL AUTOENCODERS

**Gustav Bredell, Kyriakos Flouris, Krishna Chaitanya, Ertunc Erdil & Ender Konukoglu**
Department of Information Technology and Electrical Engineering
ETH Zurich
`gustav.bredell@vision.ee.ethz.ch`

## ABSTRACT

Variational autoencoders (VAEs) are powerful generative modelling methods, however they suffer from blurry generated samples and reconstructions compared to the images they have been trained on. Significant research effort has been spent to increase the generative capabilities by creating more flexible models but often flexibility comes at the cost of higher complexity and computational cost. Several works have focused on altering the reconstruction term of the evidence lower bound (ELBO), however, often at the expense of losing the mathematical link to maximizing the likelihood of the samples under the modeled distribution. Here we propose a new formulation of the reconstruction term for the VAE that specifically penalizes the generation of blurry images while at the same time still maximizing the ELBO under the modeled distribution. We show the potential of the proposed loss on three different data sets, where it outperforms several recently proposed reconstruction losses for VAEs.

## 1 INTRODUCTION

Generative modelling aims to learn a data distribution $P_D$ from samples, such that new samples can be generated from the learned distribution, x $\sim P_D$. The learned distribution can be used for a variety of tasks ranging from out-of-distribution detection (Asim et al. (2020)) to serving as a prior for reconstruction tasks (Tezcan et al. (2019)). One generative modelling approach of particular interest is Variational Autoencoder (VAE) as introduced by Kingma & Welling (2013). This approach is particularly interesting because it yields a lower dimensional latent model, which allows generating lower dimensional representations of samples, and for doing that the model parameters are determined through directly maximizing the ELBO of the training samples. The combination of these two points makes VAEs unique compared to other generative modeling approaches (Goodfellow et al. (2014), Rezende & Mohamed (2015), Ho et al. (2020), Brehmer & Cranmer (2020), Caterini et al. (2021)).

One major drawback of VAEs is that they often produce blurry generative samples even though the images in the training distribution were sharp. This is due to the formulation of the optimization. Variational autoencoders are optimized by maximizing the evidence lower bound (ELBO) $\mathbb{E}_{z \sim q_\phi(z|x)} log[p_\theta(x|z)] - D_{KL}[q_\phi(z|x)||p(z)]$ with respect to the network parameters $\theta$ and $\phi$. The first term is often referred to as the reconstruction loss and effectively ensures that an observed sample can be mapped to the latent space through the posterior model and reconstructed back from its possible latent representation. The second term is matching the learned approximate posterior distribution, $q_\phi(z|x)$, with that of a prior distribution $p(z)$. The ELBO can also be formulated as minimizing the Kullback-Leibler divergence in the augmented space, where the data distribution is expanded with the auxiliary variable z, $D_{KL}[q_{D,\phi}(x,z)||p_\theta(x,z)]$ as described by Kingma et al. (2019). It is clear that the model is penalized heavily if samples are likely under $q_{D,\phi}(x,z)$ but not $p_\theta(x,z)$, however not the other way around due to the asymmetry of the Kullback-Leibler divergence. Therefore, $p_\theta(x,z)$ and thus $p_\theta(x)$ will have a larger variance than the original data distribution $q_{D,\phi}(x,z)$ leading to generated samples being more diverse than original data, practically also including blurry images. To alleviate this issue both $q_\phi(z|x)$ and $p_\theta(x|z)$ should be flexible enough as has been the objective of other works (Berg et al. (2018), Kingma et al. (2016), Vahdat & Kautz (2020)).

In addition to the line of work on increasing the flexibility of the posterior $q_\phi(z|x)$, which often come at the cost of being more difficult to optimize and computationally expensive, there is a line of work focusing on improving the reconstruction loss formulation. To keep the mathematical link to maximizing $p_\theta(x)$ for the observed samples through the ELBO formulation, a distribution $p_\theta(x|z)$ has to be assumed for the reconstruction loss. Popular choices for these distributions are Gaussian or Bernoulli as introduced by Kingma & Welling (2013). The Gaussian distribution, $p_\theta(x|z) = \mathcal{N}(\mu(z)_\theta, \Sigma)$ has seen widespread adaption (Castrejon et al. (2019),Lee et al. (2020)) and leads to a simplification of the reconstruction loss to the mean squared error (MSE) if $\Sigma$ is assumed to be identity. Since the latent space $z$ has a lower dimensionality than $x$, a perfect reconstruction will not be possible and the reconstruction loss will determine which features of $x$ to weigh the most. The MSE places no weight on specific features, all features will be reconstructed with a similar weighting. Since most power in natural images is located in the lower frequency in their spectrum Van der Schaaf & van Hateren (1996), low frequency features, in other words blurry features, will dominate the reconstruction loss. Consequently, reconstruction fidelity in higher frequency features will be less important.

Several previous works have recognized the potential of augmenting the reconstruction loss to address the blur, but they approached the problem implicitly. They aimed to retrieve more visually pleasing generations. One initial approach by Hou et al. (2017) was to replace the pixel-wise loss with a feature based loss that is calculated with a pre-trained convolutional neural network (CNN). However, this loss is limited to the domain of images on which the CNN was pre-trained. It has also been suggested in literature to combine generative adverserial networks with VAEs by replacing the reconstruction loss of the VAE with an adversarial loss (Larsen et al. (2016)). This approach comes at the expense of architecture changes and with optimization challenges. Another approach by Barron (2019) optimizes the shape of the loss function during training to increase the robustness. While their loss formulation has shown improved results, it was not specifically focusing on sharpening VAE generations. By learning the parameters of a Watson perceptual model and using it as a reconstruction loss, Czolbe et al. (2020) have shown that generative examples of VAEs can be improved. Since the human perception also values sharp examples, the proposed loss improved sharpness of samples, but optimization with the loss is less stable than other methods. Recently, Jiang et al. (2021) tackled the blur problem directly in the frequency domain. They have introduced the focal frequency loss, which applies a per frequency weighting for the error term in the frequency domain, where the weighting is dependent on the magnitude of the error at that frequency. The above mentioned methods do not explicitly focus on reducing blur errors, and, except for the work by Barron (2019), the proposed reconstruction terms lose their mathematical link to maximizing the ELBO, since the distribution $p_\theta(x|z)$ is not defined.

In this paper we aim to explicitly minimize the blur error of VAEs through the reconstruction term, while at the same time still perform ELBO maximization for the observed samples. We derive a loss function that explicitly weights errors that are due to blur more than other errors. In our experiments, we show that the new loss function produces sharper images than other state-of-the art reconstruction loss functions for VAEs on three different datasets.

## 2 BACKGROUND ON BLURRING AND SHARPENING

In this section we will review some background on blurring and image sharpening. The blurring degradation of a sharp image, $x$ can be modeled by convolving $x$, with a blur kernel, $k$, $\hat{x} = x * k$, where $\hat{x}$ denotes the degraded image. Assuming no additive noise and the blurring kernel does not suppress any frequencies, using the convolution theorem, the blurring operation can be inverted in the frequency domain as $\mathcal{F}(\hat{x})/\mathcal{F}(k) = \mathcal{F}(x)$, where $\mathcal{F}$ is the Fourier transform. The sharp $x$ can then be obtained by applying the inverse Fourier transform to the ratio.

The assumptions of knowing the blur kernel $k$, that no noise was present during the blur generation, and convolving with $k$ not suppressing any frequencies are strong. In the case that $k$ is not well known and $\mathcal{F}(k)$ has several terms close to zero, division by $\mathcal{F}(k)$ could cause extreme values. To provide a more stable inverse operation, one can use the Wiener deconvolution (Wiener (1949)),

$$\frac{\mathcal{F}^*(k)}{|\mathcal{F}(k)|^2 + C}\mathcal{F}(\hat{x}) \approx \mathcal{F}(x), \tag{1}$$

where $\mathcal{F}^*$ is referring to the complex conjugate and $C$ is a constant to offset $\mathcal{F}(k)$ values that are close to zero and low signal-noise-ratio (SNR). In the presence of noise, $C$ is chosen as $1/$ SNR Gonzalez & Woods (2001); Wiener (1949), or a constant.

## 3 METHOD

The motivation of our work is to reduce the blur in the generated outputs of variational autoencoders (VAEs). We propose to do this by weighting the reconstruction term to focus on errors induced by the blur. In VAEs the evidence lower bound (ELBO) is maximized for the observed samples, that are coming from the real data distribution $p(x)$. Assuming the constraint on the latent space is a zero-centered unit Gaussian, this leads to the following optimization problem, Rezende & Mohamed (2015),

$$\max_{\theta,\phi} \mathbb{E}_{z \sim q_\phi(z|x)} \log[p_\theta(x|z)] - D_{KL}[q_\phi(z|x)||\mathcal{N}(0, \mathbb{I})], \tag{2}$$

where $x$ and $z$ correspond to the input image and latent variable, respectively, and $\theta$ and $\phi$ denote the network parameters. By assuming that likelihood, $p_\theta(x|z)$ is a multi-variate Gaussian, the ELBO can be written as:

$$\min_{\theta,\phi} \mathbb{E}_{z \sim q_\phi(z|x)} \log(|\Sigma_\theta(z)|^{\frac{1}{2}}) + \frac{1}{2}(x - \hat{x}_\theta(z))^T \Sigma_\theta(z)^{-1}(x - \hat{x}_\theta(z)) + D_{KL}[q_\phi(z|x)||\mathcal{N}(0,\mathbb{I})], \tag{3}$$

where $\hat{x}_\theta(z)$ is the mean of the multi-variate Gaussian and $\Sigma_\theta(z)$ is its covariance. Generally, the covariance matrix $\Sigma_\theta(z)^{-1}$ is assumed to be the identity times a constant scalar, hence no dependence on $\theta$ nor $z$, which has the advantage that the calculation of determinant of $\Sigma_\theta(z)$ is not necessary. The covariance-matrix can also be seen as a weighting term of the reconstruction loss compared to the KL-divergence. In the next sections we derive the proposed form for the covariance matrix.

### 3.1 ERROR INDUCED BY BLUR

To design a weighting map that targets errors that are caused by the blur, those errors need to be formalized. For the sake of notational simplicity, we will drop $z$ dependence of $\hat{x}_\theta$ and $\Sigma_\theta$ for now. Let us assume that the output of the VAE is a blurred version of the input image, i.e., $\hat{x}_\theta = x * k$. Therefore, the squared reconstruction error can be viewed as the squared distance between the input and the blurred version of the input. Using Parseval's identity Parseval (1806), in Fourier space this corresponds to

$$\|\mathcal{F}(e)\|^2 = \|\mathcal{F}(x) - \mathcal{F}(x) \cdot \mathcal{F}(k)\|^2 = \|\mathcal{F}(x)(1 - \mathcal{F}(k))\|^2 = \|\mathcal{F}(x)|1 - \mathcal{F}(k)|\|^2, \tag{4}$$

where $e$ is the error term. We see that the magnitude of the blur error at any frequency is proportional to $|\mathcal{F}(x)||1 - \mathcal{F}(k)|$. The implication of this can best be understood if we imagine $k$ as a centered Gaussian blur with unit sum. Then, the Fourier transform of $k$ will also be a Gaussian with its largest value at the center of the Fourier space, i.e., $(u, v) = (0, 0)$. Moreover, because $k$ sums up to 1, the largest value of $\mathcal{F}(k)$ will be smaller than 1. Therefore, for low frequencies $(u, v) \approx (0, 0)$ the value of $\mathcal{F}(k)$ would be positive and high, yielding low $|1 - \mathcal{F}(k)|$. For higher frequencies the value of $\mathcal{F}(k)$ would be positive and low, yielding high $|1 - \mathcal{F}(k)|$. This looks like the major contribution of blur error is due to higher frequencies. So, minimizing for the squared error, should directly focus on removing the blur. However, it is crucial to note that analyses on the power spectra of natural images showed that the power of natural images drops with increasing frequency with the power law $1/w^\alpha$ with $\alpha \approx 2$ Van der Schaaf & van Hateren (1996), where $w$ denotes an arbitrary convex combination of $u$ and $v$. Thus, even though the multiplicative factor $|1 - \mathcal{F}(k)|$ is larger for higher frequencies, because image's power at higher frequencies is lower than at lower frequencies, their contributions to $|\mathcal{F}(e)|^2$ will still be smaller compared to lower frequencies. Hence, reconstruction term will be dominated by the lower frequency errors. As a result, models obtained by minimizing for $\|F(e)\|$ will likely tolerate blurry reconstructions and generations.

### 3.2 WEIGHING THE ERROR THROUGH WIENER DECONVOLUTION

Given the distribution of the blur induced errors, the next question to answer is how to best weigh the error term to emphasize blur errors due to the convolution with a $k$. If $k$ is known, one way of

weighting the errors would be with the inverse value of $\mathcal{F}(k)$. For small blur errors with $\mathcal{F}(k) \approx 1$ the weighting would be $1$ and does not change the error term. However, as $|\mathcal{F}(k)|$ gets smaller the error weighting will increase. With any weight matrix in the Frequency domain, one can compute a weighted reconstruction error as

$$\|\mathcal{W} \cdot (\mathcal{F}(x) - \mathcal{F}(\hat{x}_\theta))\|^2, \tag{5}$$

where $\mathcal{W}$ is a chosen weight matrix. To specifically weigh the blur induced errors higher, we would like to chose $\mathcal{W} = \frac{1}{\mathcal{F}(k)}$, such that

$$\|\frac{1}{\mathcal{F}(k)} \cdot (\mathcal{F}(x) - \mathcal{F}(\hat{x}_\theta))\|^2. \tag{6}$$

As mentioned before, this naive deconvolution of the reconstruction term, would lead to instabilities for frequencies where $\mathcal{F}(k) \approx 0$. To stabilize the chosen weighting method, we follow the Wiener deconvolution and add a constant $C$ as

$$\|\frac{\mathcal{F}^*(k)}{|\mathcal{F}(k)|^2 + C} \cdot (\mathcal{F}(x) - \mathcal{F}(\hat{x}_\theta))\|^2. \tag{7}$$

Linking this to the VAE, a reconstruction loss in the Fourier transform corresponds to a Gaussian likelihood $p_\theta(x|z)$ with $\Sigma^{-1} = W^T W$, where $W$ corresponds to the matrix that encodes the convolution operation defined by the inverse Fourier transform of $\mathcal{W}$, i.e., $\mathcal{F}^{-1}(\mathcal{W})$. When $\mathcal{W}$ is defined in terms of the kernel $k$ as in Equation 7, then we denote this dependence with a subscript, i.e., $\Sigma_k^{-1} = W_k^T W_k$ with $W_k$ being the matrix corresponding to convolution with $\mathcal{F}^{-1}(\frac{\mathcal{F}^*(k)}{|\mathcal{F}(k)|^2 + C})$. As a result, using the Parseval's identity the $\log p_\theta(x|z)$ term needed for the computation and optimization of the ELBO can be written as

$$\log p_\theta(x|z) = -\frac{D}{2}\log(2\pi) - \frac{1}{2}\log|\Sigma_k| - \frac{1}{2}\left\|\frac{\mathcal{F}^*(k)}{|\mathcal{F}(k)|^2 + C}(\mathcal{F}(x) - \mathcal{F}(\hat{x}_\theta))\right\|^2. \tag{8}$$

### 3.3 Determining the Blur Kernel k

To determine the weight matrix at a given iteration during optimization, we need a blur kernel $k$. To determine $k$ we utilize the assumption of image degradation via a convolution with a blur kernel as in Section 2. Specifically, we optimize to find the kernel $k$ that minimizes the means squared error when the input $x$ is blurred with kernel $k$ with the output of the VAE, $\hat{x}_\theta$,

$$\min_k \|x * k - \hat{x}_\theta\|^2. \tag{9}$$

It is reasonable to assume that some images might be blurred more than others by the VAE, and we confirm this in Section 4.3, and it changes with the parameters of the networks. Therefore, $k$ would have to be determined through the optimization after every iteration and for every image. To this end, we utilize a neural network, $G$, that takes as input $z$ sample from $q(z|x)$ and outputs a kernel $k$. The optimization can consequently be written as

$$\min_\gamma \|x * G_\gamma(z) - \hat{x}_\theta\|^2, \tag{10}$$

where $k = G_\gamma(z)$ and $z \sim \mathcal{N}(0, I)$.

### 3.4 Determinant of Covariance Matrix

The non-trivial choice of $\Sigma$ for this method, implies that the determinant $|\Sigma|$ needs to be accounted for in the likelihood calculation, Equation 8. Full computations of determinants are costly, nevertheless, matrices defined by convolution operations provide special opportunities. The convolution operation can be implemented as a matrix multiplication with an appropriate circulant matrix $K$ as $\text{vec}(k * x) = K \cdot \text{vec}(x)$, where $\text{vec}(\cdot)$ is the vectorization operation. If the kernel $k$ is two dimensional this will result in a block-circulant matrix with each block being a circulant matrix as well. Circulant and block-circulant matrices have the property that their determinant can be calculated analytically as shown by Davis (1979). In addition, the circulant matrices will be sparse if the size of $k$ is much smaller than that of $x$, which enables an efficient computation of the determinant.

In our case $\Sigma^{-1}$ is equal to $W_k^T W_k$, where $W_k$ represents the block-circulant matrix that corresponds to the Wiener deconvolution constructed with the kernel $k$. If we assume the constant $C$ is zero and $k$ does not suppress any frequencies for any $x$, then the deconvolution would be the inverse operation of the convolution, i.e., $\mathcal{W}_k = 1/F(k)$, and we can write $\mathcal{F}^{-1}(\mathcal{W}_k \mathcal{F}(k)\mathcal{F}(x)) = x, \forall x$. This implies $W_k \cdot \mathrm{vec}(k * x) = \mathrm{vec}(x), \forall x$ and $W_k \cdot K \cdot \mathrm{vec}(x) = \mathrm{vec}(x), \forall x$ with $K$ being the block circulant matrix that corresponds to the convolution with $k$. Therefore, $W_k = K^{-1}$ and $\Sigma_k = (W_k^T W_k)^{-1} = KK^T$. Hence, the determinant of the covariance can be computed in terms of the determinant of $K$ as $|\Sigma_k| = 2|K|$. Using the properties of block-circulant matrices, this determinant can be easily computed. However, any blurring $k$ will likely suppress some frequencies and therefore for optimization purposes $C$ has to be set to a small positive constant to avoid division by zero. The inverse of $W_k$ is then the block-circulant matrix that corresponds to $\frac{|\mathcal{F}(k)|^2 + C}{\mathcal{F}^*(k)} = \mathcal{F}(k) + \frac{C}{\mathcal{F}^*(k)}$. $C/\mathcal{F}^*(k)$ can not be explicitly computed. Instead, we approximate the inverse of $W_k$ as $K(\epsilon) = K + \epsilon \mathbb{I}$, with $K$ still being the block-circulant matrix corresponding to $k$. For $k$ that suppresses large portion of the Fourier space, this approximation will behave similar to Wiener deconvolution in the sense that it will not allow suppression of frequencies completely since the Fourier transform of the kernel that corresponds to $K(\epsilon)$ will never be zero.

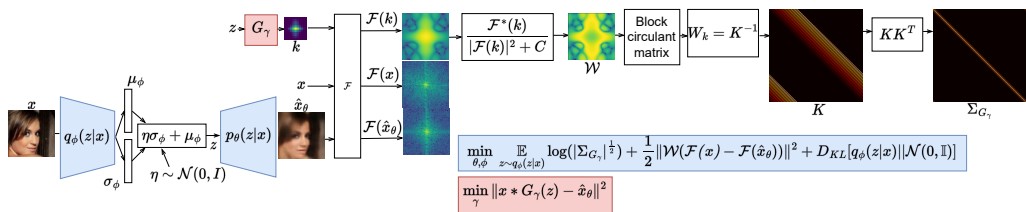

Figure 1: Sketch of the proposed approach for minimizing the blur error in VAEs. We illustrate the two losses that we minimize in the alternating optimization and the corresponding network parameters that are updated in different colors.

## 3.5 VAE OPTIMIZATION

In the previous section the dependency of $\Sigma$ on $z$ has been omitted for notational simplicity. We showed the inverse of $W_k$ can be approximated with $K + \epsilon \mathbb{I}$ that would allow an estimate of the determinant of $\Sigma$. From Equation 10, we also saw that the kernel $k$ that constructs $K$ is dependent on $z$ and is modeled with a neural network with the parameters $G_\gamma(z)$. Thus, the determinant will be both influenced by $z$ and $\epsilon$. If we assume $\epsilon$ to be small, then the determinant will be mainly dependent on $z$. However, if we assume $\epsilon$ to be large, then the identity will dominate the determinant calculation and the influence of $z$ on the determinant will negligible. Applying the reconstruction term derived in section 3.2 and estimating the determinant of $\Sigma$ through $K(G_\gamma(z), \epsilon)$ leads to the following optimization equations:

$$\max_{\theta, \phi} \log |(K(G_\gamma(z), \epsilon)| + \frac{1}{2}\| \frac{\mathcal{F}^*[G_\gamma(z)]}{|\mathcal{F}[G_\gamma(z)]|^2 + C} \cdot (\mathcal{F}(x) - \mathcal{F}(\hat{x}_\theta))\|^2 - D_{KL}[q_\phi(z|x)||\mathcal{N}(0, \mathbb{I})],$$

(11)

$$\min_{\gamma} \|x * G_\gamma(z) - \hat{x}_\theta\|^2,$$

(12)

where alternating optimization is used. We note that after each iteration of Equation 12 the model for which we optimize the ELBO in Equation 11, changes. The update of the model is incremental though due to the alternating optimization and in practice no convergence issues were observed. We illustrate a sketch of the proposed approach in Figure 1.

## 4 EXPERIMENTS

To validate the contribution of the proposed approach we perform experiments on two natural image datasets and one medical image dataset. We first provide a dataset and experimental implemen-

tation description followed by an ablation study to validate the assumptions made in the method section. Finally, we provide quantative and qualitative results for our approach compared to other reconstruction terms used in literature.

## 4.1 DATASETS

To evaluate the potential of the proposed approach on natural images we make use of the popular CelebA dataset as provided by Liu et al. (2015) and Lee et al. (2020) for low and high-resolution images, respectively. For the low-resolution images we center-crop the images to $64 \times 64$ and for the high-resolution we use $256 \times 256$. The low-resolution and high-resolution datasets consists of 200'000 and 30'000 images, respectively. Furthermore, to investigate the performance of the proposed method beyond natural images we utilize the Human Connectome Project (HCP) dataset (Van Essen et al. (2013)) and use the same preprocessing as Volokitin et al. (2020). Of the 960 volumes with a $256 \times 256 \times 256$ we utilize 400 volumes, from which we randomly crop patches of size $64 \times 64$. For all the datasets we use 80% for the training set and 20% for the test set.

## 4.2 EXPERIMENTAL SETTINGS

The code is written in Python and PyTorch (Paszke et al. (2019)) is used as library for the deep learning models. The architecture of the VAE consisted of $N$ convolutional encoding layers and decoding layers, that is connected through a multilayer perceptron which does the final compression and initial of scaling up of the latent variable with the size of 256. Details on the architecture can be found in the code that will be publicly available upon acceptance. Furthermore, the Adam optimizer is used with a learning rate of $1e^{-4}$. For the low- and high-resolution CelebA the number of training epochs were, 100 and 200, respectively. For the HCP dataset the models were trained for 400 epochs.

As metrics we utilized the peak-signal to noise ratio (PSNR), structural similarity meassure index (SSIM), learned perceptual image patch similarity (LPIPS, Zhang et al. (2018)) and inception distance (FID, Heusel et al. (2017)) for comparing the reconstruction of the VAEs with the respective input. To quantitatively evaluate the generative examples we use the FID score and provide visual examples.

In section 3.1 we state that we assume the current reconstruction of the VAE to be a blurred version of the output, $\hat{x}_\theta = x * k$. This assumption does not hold in the first few epochs of training. Thus, we set $\Sigma^{-1}$ to $\mathbb{I}$ for the first ten and twenty epochs for the CelebA and HCP datasets, respectively. After this initial period $\Sigma^{-1}$ becomes a function of $G_\gamma(z)$. To avoid $G_\gamma(z)$ initially producing inaccurate estimates of the blur kernel we already start training $G_\gamma(z)$ even when $\Sigma^{-1}$ is still set to $\mathbb{I}$. This allows accurate blur kernel prediction when $\Sigma^{-1}$ becomes a function of $G_\gamma(z)$.

## 4.3 ABLATION STUDY

To obtain a better understanding of the influence of the different parameters of the proposed approach we ran an evaluation on the low-resolution CelebA dataset as shown in Table 1. We focused on two aspects of the current approach. Firstly, we investigated how the choice of $\Sigma$ influences the performance, by comparing a fixed $\Sigma$ that is not dependent on the latent variable $z$ with that of a $\Sigma$ that is a function of $z$ but with a varying size $\epsilon$. We modeled the $\Sigma$ that is not a function of $z$ as a Gaussian kernel with a with a fixed radius. This corresponds to a fixed weighting term in the reconstruction loss and a constant determinant, which can thus be omitted from the optimization. We found that this approach outperformed the configuration where $\Sigma$ is a function of $z$ with a small $\epsilon$. During optimization the latent space is adjusted to minimize the $log|\Sigma|$ and even though this normally cause $\Sigma^{-1}$ to increase the error term significantly, the constant $C$ prevents this term from very large values. The best performance was achieved by assuming $\epsilon$ to be large. A large $\epsilon$ would dominate $log|\Sigma|$ and the influence of the value of $G_\gamma(z)$ on the determinant would be minimal, allowing us to omit optimizing over the determinant by assuming it is roughly constant. The second parameter we examined was the constant $C$ in the Wiener Filter by changing it value for the scenario where $\Sigma$ is a function of $z$ and $\epsilon$ is large. We observe the choice of $C$ to produce robust results for small values $\leq 0.025$. However, performance degrades with larger $C$ values.

In addition to the ablation study, we visualized the evolution of the kernel estimated by $G_\gamma(z)$ during the course of training for four example images, see Figure 2. As explained in section 4.2, $\Sigma$

| Parameter | Value | PSNR | SSIM | LPIPS | $FID_{Recon}$ | $FID_{Gen}$ |
|---|---|---|---|---|---|---|
| $\Sigma(K)$ | $K(\mathcal{N}(0,2),\epsilon)$ | 22.37 | 0.6306 | 0.1217 | 0.0616 | 0.0753 |
| | $K(\mathcal{N}(0,4),\epsilon)$ | 22.56 | 0.646 | 0.1097 | 0.0615 | 0.0823 |
| | $K(G_\gamma(z),\epsilon_{small})$ | 19.51 | 0.4463 | 0.1924 | 0.171 | 0.214 |
| | $K(G_\gamma(z),\epsilon_{large})$ | 23.21 | 0.7296 | 0.1254 | 0.0364 | 0.0536 |
| **C** | 0.005 | 23.21 | 0.7294 | 0.1186 | 0.0340 | 0.0556 |
| | 0.025 | 23.21 | 0.7296 | 0.1254 | 0.0364 | 0.0536 |
| | 0.1 | 22.87 | 0.7127 | 0.1423 | 0.0454 | 0.0618 |

Table 1: We investigate the performance differences for different modelling choices of the blur kernel estimation, $G_\gamma(z)$ versus constant $\mathcal{N}(0,var)$. In addition, we look at the influence of $\epsilon$ on the optimization. We observe a kernel generator that is dependent on $z$, $G_\gamma(z)$, to perform the best if $\epsilon$ is assumed to be large. Under this condition we also examine the influence of the constant $C$ in the Wiener Filter.

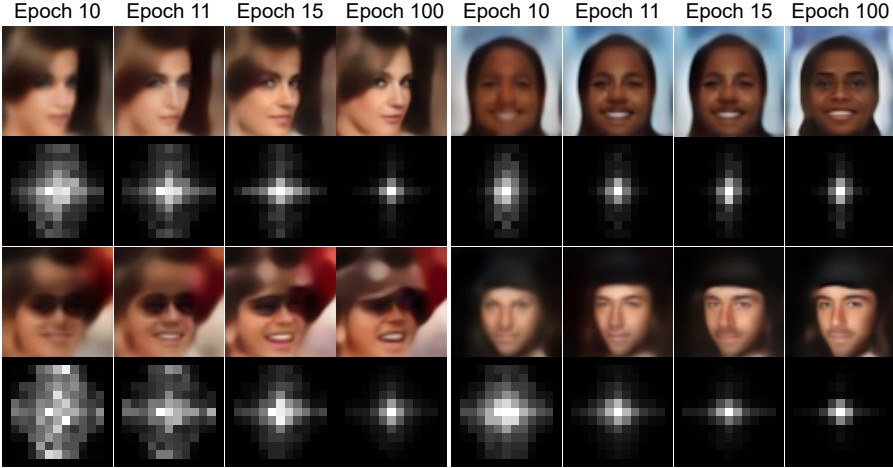

Figure 2: The evolution of the estimated kernel, $G_\gamma(z)$, is shown four different different images at different epochs during training. The blur minimizing reconstruction term is introduced at epoch 10, after which a strong decrease of the estimated blur kernel can be observed along with sharper image reconstruction. In addition, it can be seen that different images have different blur kernel estimates, which motivates determining the blur kernel per image and making $\Sigma$ dependent on $z$.

is initially modeled for ten epochs by an normal multivariate Gaussian until the assumption holds that the output can be seen as a blurred version of the input. It can be seen that the estimated blur kernel is large for all four images and in the subsequent epochs (11 and 15) drastically decreases as we are specifically penalizing blurry outputs. This minimization of the blur continues till the end of training at iteration 100, where it can be observed that the blur in the image as well as the corresponding estimated blur kernel is much smaller than at the start. We note that a residual blur remains due to the inherent reconstruction limitations of a VAE such as the information bottleneck by constructing a lower dimensional latent space. Another observation is that the estimated blur kernel is different for each of the four images. This provides some empirical evidence that it is a good idea to model $\Sigma$ as a function of $z$ to capture this dependency. This also provides an explanation as to why $\Sigma(z)$ outperformed a fixed $\Sigma$ as seen in the ablation study.

## 4.4 QUANTITATIVE AND QUALITATIVE RESULTS

We compare our proposed method with several reconstruction terms that are often used in literature on three different datasets. For the learned perceptual losses, namely the Watson perceptual loss (Czolbe et al. (2020)) and VGG-Loss (Zhang et al. (2018)) we use the implementation provided by Czolbe et al. (2020), where the losses were learned for with the Two-Alternative Forced Choice

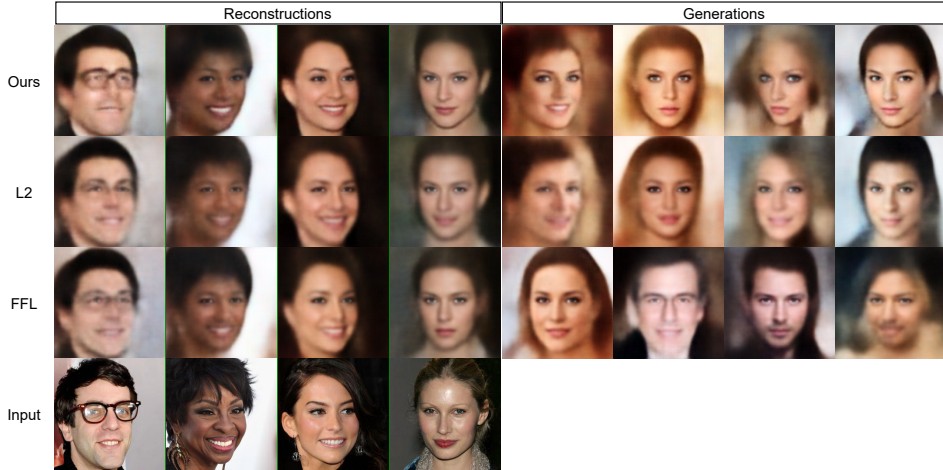

Figure 3: Here, we present the qualitative results of reconstructions and generations from the proposed method and relevant compared methods on CelebA256 dataset. We observe lower blurriness and higher sharpness for the reconstructed images from the proposed method.

| Dataset | Method | PSNR | SSIM | LPIPS | $FID_{Recon}$ | $FID_{Gen}$ |
|---|---|---|---|---|---|---|
| **CelebA** | L2 | 22.51±1.6 | 0.6944±6.9$e^{-2}$ | 0.1656±6.4$e^{-2}$ | 0.0571 | 0.0656 |
| | L1 | 22.68±1.9 | 0.7069±7.1$e^{-2}$ | 0.176±6.4$e^{-2}$ | 0.0671 | 0.0725 |
| | SSIM | 20.79±1.8 | 0.7248±6.5$e^{-2}$ | 0.1691±7.2$e^{-2}$ | 0.0487 | 0.0592 |
| | CE | 22.95±1.7 | 0.7183±6.9$e^{-2}$ | 0.148±6.4$e^{-2}$ | 0.045 | 0.0575 |
| | FFL | 23.02±1.6 | 0.7141±6.5$e^{-2}$ | 0.1552±6.1$e^{-2}$ | 0.056 | 0.0642 |
| | **Ours** | **23.21±1.6** | **0.7296±6.3$e^{-2}$** | **0.1254±5.8$e^{-2}$** | **0.0364** | **0.0536** |
| | VGG | 17.64±1.7 | 0.4439±8.9$e^{-2}$ | 0.0934±4.0$e^{-2}$ | 0.0282 | 0.0363 |
| | Watson* | 19.22±2.1 | 0.5808±1.1$e^{-1}$ | 0.1164±5.5$e^{-2}$ | 0.0356 | 0.0459 |
| **CelebA-HQ** | L2 | 19.37±1.4 | 0.5632±7.9$e^{-2}$ | 0.5372±7.6$e^{-2}$ | 0.1123 | 0.1138 |
| | L1 | 20.49±1.6 | **0.6027±7.9$e^{-2}$** | 0.5132±8.2$e^{-2}$ | 0.1046 | 0.1038 |
| | SSIM | 17.99±1.8 | 0.5953±8.1$e^{-2}$ | 0.561±8.0$e^{-2}$ | 0.1065 | 0.0982 |
| | CE | **20.54±1.6** | 0.5958±7.8$e^{-2}$ | 0.4922±7.9$e^{-2}$ | 0.0909 | 0.1053 |
| | FFL | 20.15±1.4 | 0.5779±7.7$e^{-2}$ | 0.5179±7.6$e^{-2}$ | 0.1068 | 0.1124 |
| | **Ours** | 20.17±1.5 | 0.5867±7.8$e^{-2}$ | **0.4874±8.0$e^{-2}$** | **0.0858** | **0.0938** |
| | VGG | 11.61±1.6 | 0.3462±6.5$e^{-2}$ | 0.4943±5.5$e^{-2}$ | 0.1237 | 0.1231 |
| | Watson | 12.68±4.6 | 0.4869±7.5$e^{-2}$ | 0.5601±8.0$e^{-2}$ | 0.1295 | 0.1381 |

Table 2: We provide quantative results (average±std) on the low ($64 \times 64$) and high quality ($256 \times 256$) CelebA dataset. L1, L2 and CE refer to the first norm, second norm and cross-entropy, respectively, as losses. Learned reconstruction losses are separated with a line. We observe our method to perform well across all metrics for both datasets.

dataset (Zhang et al. (2018)). For the focal frequency loss (FFL) we used the code the authors provided (Jiang et al. (2021)). For all the losses we use the same network architecture and training parameters except for the Watson loss on CelebA, where we used the authors model, which has a factor of three more parameters than the model used for the other methods, due to instability during training with our model architecture. We tuned the weighting between the two terms in the ELBO for all methods (Higgins et al. (2016)) and reported results where the FID score for the generative samples were minimized to ensure optimal visual performance. In Table 2, we present the results for the CelebA dataset for both low ($64 \times 64$) and high resolution ($256 \times 256$) images. It can be seen that our approach performs well across all metrics. For some metrics such as the PSNR or SSIM of the reconstruction, we get slightly smaller values, but achieve higher scores on the perceptual metrics, including generation. The perceptual losses perform very well on the perceptual metrics

| Dataset | Method | PSNR | SSIM |
|---------|--------|------|------|
| **HCP** | L2 | 18.09±5.7 | 0.4313±$1.7e^{-1}$ |
|  | L1 | 18.95±6.0 | 0.5005±$1.9e^{-1}$ |
|  | SSIM | 18.06±5.8 | **0.5922±2.1**$e^{-1}$ |
|  | Cross-Entropy | 18.69±6.0 | 0.4708±$1.8e^{-1}$ |
|  | FFL | 18.88±5.9 | 0.4795±$1.8e^{-1}$ |
|  | **Ours** | **19.54±6.1** | 0.5209±$1.9e^{-1}$ |

Table 3: We provide quantative results (average±std) on the HCP dataset, where we trained the VAE on random crops with a resolution of $64 \times 64$. It can be seen that our proposed method can easily be applied to non-natural image, while outperforming several established reconstruction losses.

for the CelebA dataset, however these metrics are often high despite obvious artifacts. In addition, the perceptual losses did not allow for a stable optmization for the high resolution CelebA images, where our loss was able to scale up without any modifications, except increasing the kernel size from $11 \times 11$ to $41 \times 41$ to account for larger possible blur kernel estimates. In addition to the numerical results, we provide examples of the reconstruction and generated samples for the high quality CelebA in Figure 3 and a larger Figure 7.

The advantage of our proposed method is that it is not constrained to the natural image domain and does not need to be re-trained for each domain. We show the potential of using our approach in the medical image analysis field by training a VAE on MRI brain slices. Since the perceptual metrics can not be applied here, we only report the MSE and SSIM when compared against other methods in Table 3. It can be seen that our approach works well in other domains and outperforms several other reconstruction terms without needing to modify our approach. We also provide some qualitative examples in Figure 4 and more examples in Figure 9 in the Appendix.

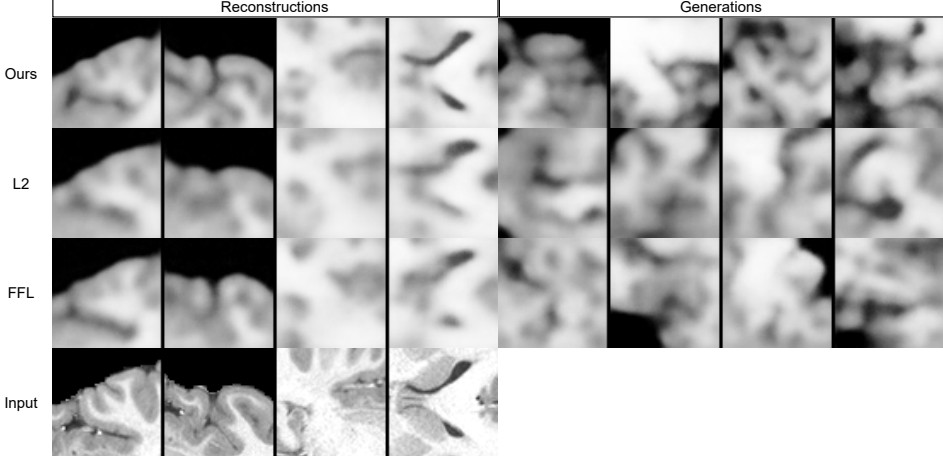

Figure 4: Here, we present the qualitative results of reconstructions and generations from the proposed method and relevant compared methods on HCP medical dataset. We observe lower blurriness and higher sharpness for the reconstructed images from the proposed method.

## 5 CONCLUSIONS

In this paper, we have proposed a reconstruction term, which minimizes the blur error of VAEs, and is simultaneously consistent with the ELBO maximization. Deblurring is achieved by explicitly shifting the weight importance of the blur errors. The method implies that the likelihood is a multi-variate Gaussian, where the covariance is not assumed to be trivially the identity. The resulting calculation of the determinant of the specific covariance can be computed with remarkable efficiency and we also show that it can be omitted under some assumptions leading to a strong performance. Therefore, despite the additional computational costs, the proposed method retains the desired celerity of VAEs. We perform extensive experimentation on three different datasets, that showcase sharper images as produced by the new loss function in comparison to other state-of-the art reconstruction losses.

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

# A    ARCHITECTURE DETAILS

In our main paper we worked with one single architecture to keep settings consistent across multiple different datasets. The encoder of the VAE, $q_\phi(z|x)$, consists of four and six downscaling blocks for $64 \times 64$ and $256 \times 256$ input, respectively. The downscaling blocks consists of a 2D convolution (kernel size $= 3$, stride $= 2$) followed with a batch norm (Zhang et al. (2018)) and Leaky-Relu. The output of the downscaling blocks is then passed through a two-layer multipectron to obtain the $\mu$ and $\sigma$. The sampled latent variable $z$ is then used as input for the decoder, $p_\theta(x|z)$, which first passes the $z$ through a two linear layers and then through four decoding blocks consisting of a 2D transposed convolution (kernel size $= 4$, stride $= 2$), batch-norm and Leaky-Relu. Afterwards the output is passed through one more 2D convolution (kernel size $= 3$, stride $= 1$) and a tanh activation function to obtain the final output. The number of feature layers are [64, 128, 256, 512] for each encoding block, respectively. For the decoding block, the list of feature layer numbers is reversed. To generate the kernel $k$ from the latent variable $z$ we trained a neural network, $G_\gamma(z)$. The network consists of two linear layers where the number of hidden nodes is 1000, the input is $z$ and the output is the size of the desired kernel $k$.

To ensure that are experiments are not fine-tuned to a single model configuration, we also evaluated our method with a second architecture on the CelebA64 dataset. The results can be seen in Table 4. We observe that the same trends holds in this evaluation as in Table 2. This second architecture, $q_\phi(z|x)$, consists of five encoding blocks (2D convolution (kernel size $= 3$, stride $= 2$), batch norm, Leaky-Relu) and one linear layer that outputs $\mu$ and $\sigma$. The decoder consisted of one linear layer followed by five decoding blocks (bi-linear upsampling, 2D convolution (kernel size $= 3$, stride $= 1$), batch norm and Leaky-Relu. Afterwards the output is passed through a 2D convolution (kernel size $= 3$, stride $= 1$) and a tanh activation function to obtain the final output. The number of feature layers are [32, 64, 128, 256, 512] for each encoding block, respectively. For the decoding block, the list of feature layer numbers is reversed.

| Dataset | Method | PSNR | SSIM | LPIPS | $FID_{Recon}$ | $FID_{Gen}$ |
|---------|--------|------|------|-------|---------------|-------------|
| **CelebA** | L2 | $21.80\pm1.7$ | $0.6609\pm8.0e^{-2}$ | $0.1895\pm7.7e^{-2}$ | 0.0696 | 0.0774 |
| | L1 | $21.87\pm1.9$ | $0.6645\pm8.0e^{-2}$ | $0.1876\pm7.5e^{-2}$ | 0.0714 | 0.0742 |
| | SSIM | $19.91\pm1.9$ | $0.6725\pm8.1e^{-2}$ | $0.2026\pm8.1e^{-2}$ | 0.0747 | 0.0721 |
| | FFL | $22.28\pm1.7$ | $0.6795\pm7.5e^{-2}$ | $0.1803\pm7.1e^{-2}$ | 0.0638 | 0.0772 |
| | **Ours** | $\mathbf{22.36\pm1.8}$ | $\mathbf{0.6951\pm7.6}e^{-2}$ | $\mathbf{0.1515\pm6.8}e^{-2}$ | **0.0448** | **0.0563** |
| | VGG | $17.10\pm1.9$ | $0.4837\pm1.0e^{-1}$ | $0.0971\pm4.5e^{-2}$ | 0.0192 | 0.0330 |

Table 4: We provide quantative results on the low ($64 \times 64$) CelebA dataset with a smaller architecture than in the main paper to show that our results can are not sensitive to architecture design. L1 and L2 refer to the first and second norm as losses. We observe our method to perform well across all metrics.

# B    ADDITIONAL RESULTS

In this section we provide additional results that are complementary to the results in the main paper. Firstly, we ran out method and several baselines on the CIFAR10 dataset (Krizhevsky et al. (2009)) and report quantitative results in Table 5. It can be seen that our method performs the best across all metrics except for the FID of the generated samples. Although other methods have a lower FID score for the generated samples, this is not reflected in the visual examples provided in Figure 5. In fact, our approach seems to be better able to reconstruct details in the generative samples. This observation also holds for the reconstruction results and is supported in the strong perceptual metric results, such as LPIPS, compared to other methods. Secondly, we provide a comparison of the ELBO value for our method with that of an L2-loss on the CelebA64 dataset. The results can be seen in Table 6. We can see that our ELBO value is lower than that of when assuming a multi-variate Gaussian with a fixed co-variance matrix. To obtain the ELBO results for our method we assumed an $\epsilon$ of 0.5. Lastly, we also extend the qualitative results of the main paper to all datasets and baselines as can be seen Figures 7, 8 and 9 for the CelebA256, CelebA64 and HCP datasets, respectively.

| Dataset | Method | PSNR | SSIM | LPIPS | $FID_{Recon}$ | $FID_{Gen}$ |
|---------|--------|------|------|-------|---------------|-------------|
| **CIFAR10** | L2 | 20.43±1.8 | 0.5757±1.0$e^{-1}$ | 0.2163±6.6$e^{-2}$ | 0.0595 | 0.0833 |
| | FFL | 20.12±1.8 | 0.5517±1.1$e^{-1}$ | 0.2208±6.7$e^{-2}$ | 0.0568 | 0.0707 |
| | CE | 20.17±1.8 | 0.5560±1.1$e^{-1}$ | 0.2275±7.6$e^{-2}$ | 0.0490 | **0.0618** |
| | **Ours** | **20.7±1.6** | **0.6382±9.6$e^{-2}$** | **0.1298±4.9$e^{-2}$** | **0.0465** | 0.0829 |

Table 5: We provide quantative results (average±std) on the CIFAR10 dataset (Krizhevsky et al. (2009)). We observe our method to perform well across all metrics.

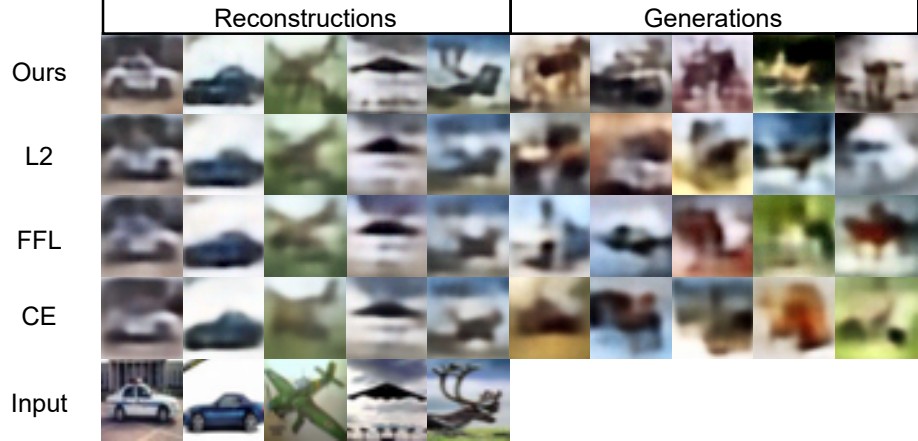

Figure 5: Here, we present the qualitative results of reconstructions and generations from the proposed method and relevant compared methods on CIFAR10 dataset (Krizhevsky et al. (2009)). We observe lower blurriness and higher sharpness for the reconstructed images from the proposed method.

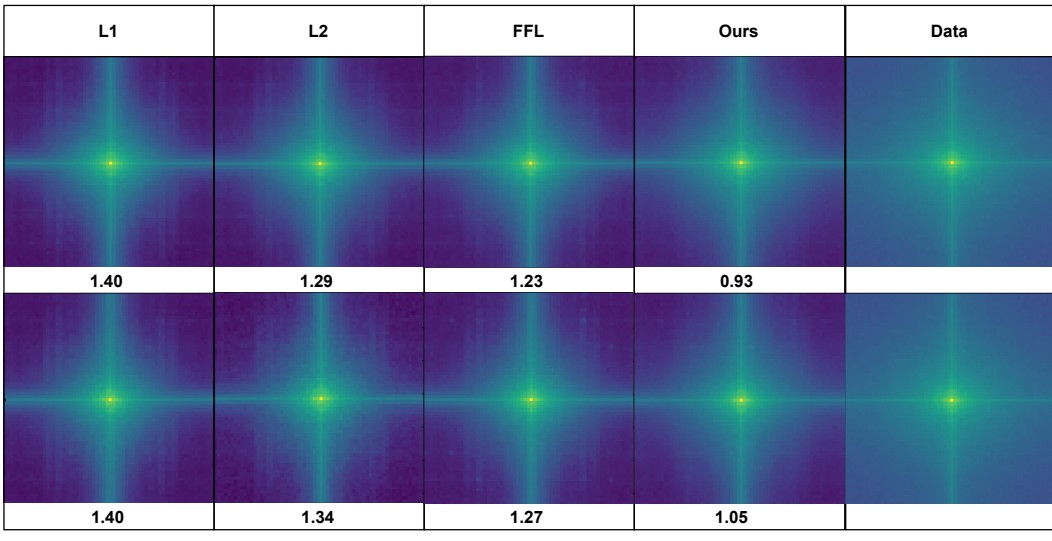

Figure 6: Here, we present the absolute value of the Fourier transform in log scale for the reconstructions and samples in the top and bottom row, respectively. To obtain the shown images we average the values over 128 individual images from the CelebA64 dataset. To the right the corresponding image for the data distribution is shown. In addition, we report the mean difference between each shown image with respect to the data image. Both qualitatively and quantitatively our approach can better replicate the frequency space of the data distribution.

| Dataset | Method | PSNR | SSIM | LPIPS | $FID_{Recon}$ | $FID_{Gen}$ | -log p |
|---------|--------|------|------|-------|-----------|---------|--------|
| **CelebA** | L2 | $22.5\pm1.6$ | $0.694\pm6.9e^{-2}$ | $0.166\pm6.4e^{-2}$ | 0.057 | 0.066 | <632 |
| | **Ours** | $\mathbf{23.2\pm1.6}$ | $\mathbf{0.730\pm6.3e^{-2}}$ | $\mathbf{0.125\pm5.8e^{-2}}$ | **0.036** | **0.054** | **<-3062** |

Table 6: We provide quantitative results on the CelebA64 dataset including the ELBO results for single image samples. For our method we assumed $\epsilon$ to be 0.5

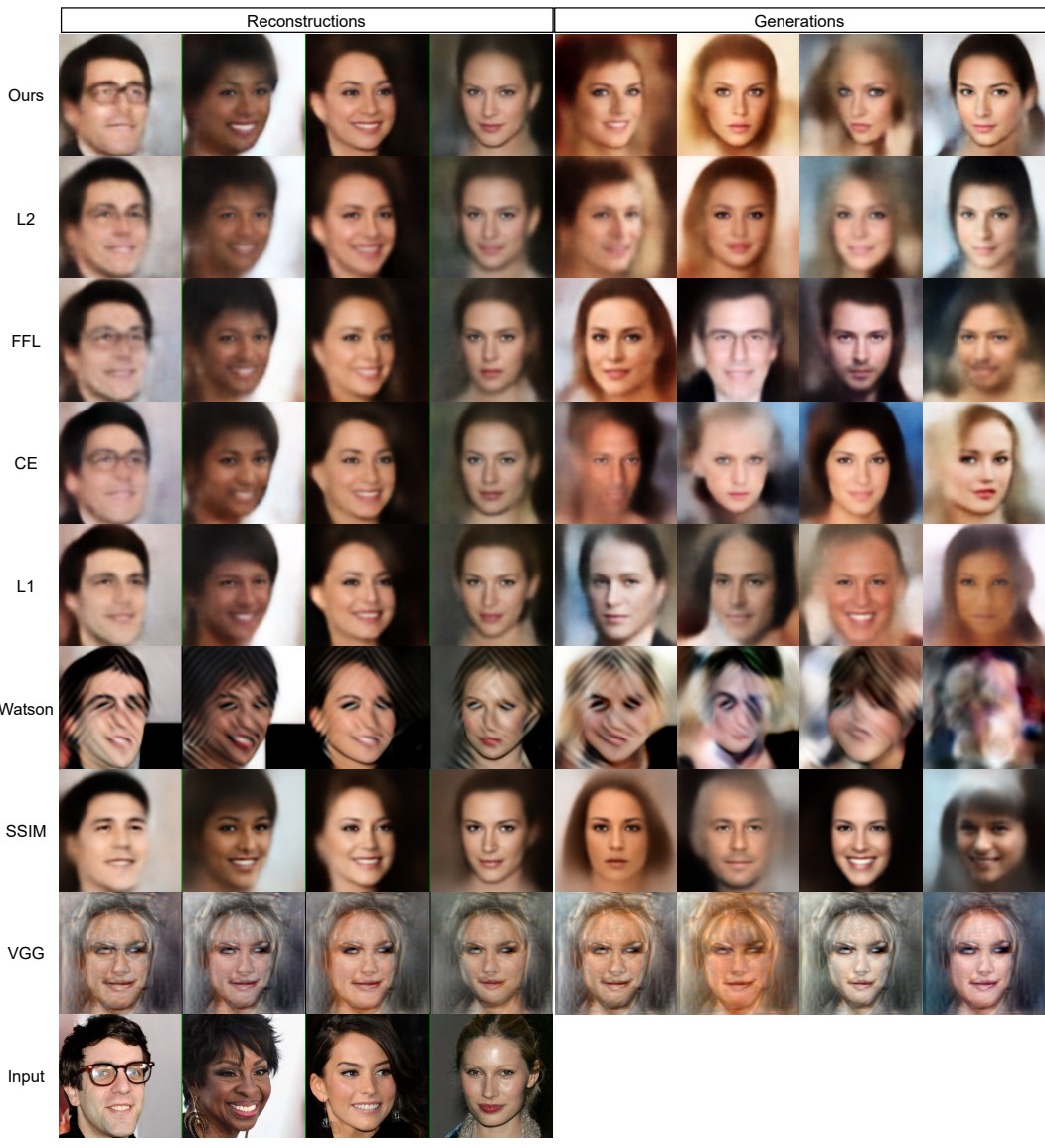

Figure 7: Here, we present the qualitative results of reconstructions and generations from the proposed method and relevant compared methods on CelebA256 dataset. We observe lower blurriness and higher sharpness for the reconstructed images from the proposed method.

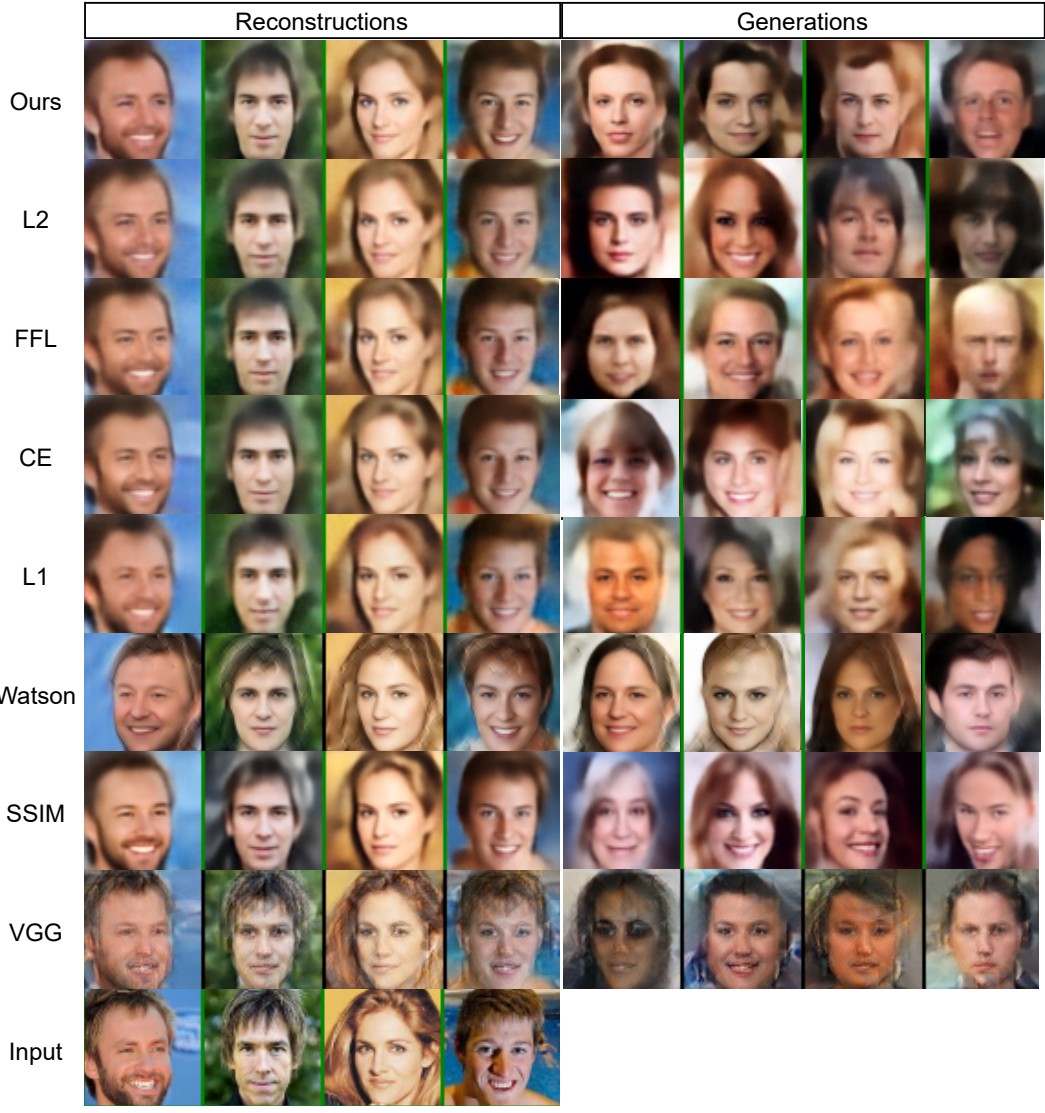

Figure 8: Here, we present the qualitative results of reconstructions and generations from the proposed method and relevant compared methods on CelebA64 dataset. We observe lower blurriness and higher sharpness for the reconstructed images from the proposed method.

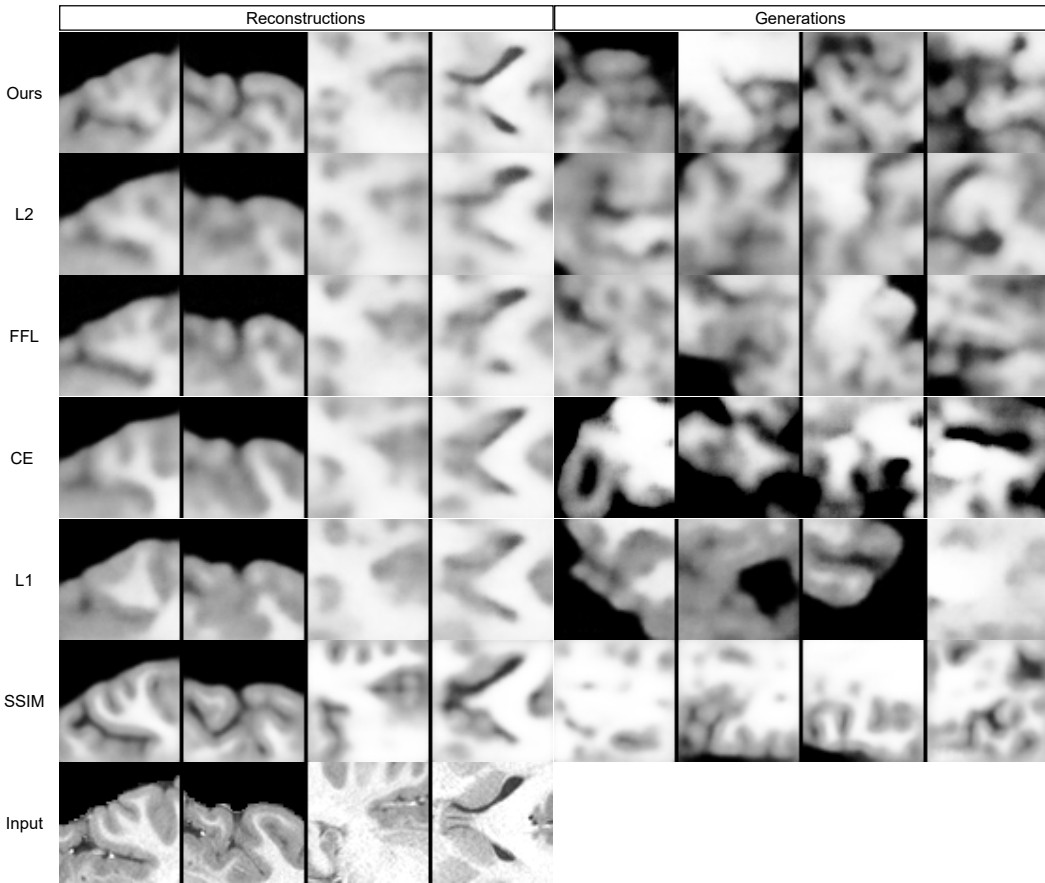

Figure 9: Here, we present the qualitative results of reconstructions and generations from the proposed method and relevant compared methods on HCP medical dataset. We observe lower blurriness and higher sharpness for the reconstructed images from the proposed method.

