# OpenReview forum: "Explicitly Minimizing the Blur Error of Variational Autoencoders"
_ICLR.cc/2023/Conference — ICLR 2023 poster_

### Official Review · Reviewer_q8AD · 2022-10-23

**Confidence:** 4
**Correctness:** 4
**Technical Novelty And Significance:** 3
**Empirical Novelty And Significance:** 3
**Recommendation:** 8

**Clarity, Quality, Novelty And Reproducibility:**

**Clarity** - For the most part, the paper is very clear in describing the motivation, proposed approach, and empirical results. The authors clearly define all mathematical terms, first walking through the simple cases, then discussing more nuanced points. The diagram of the model/method clearly lays out the basic ideas, showing both the model, a diagram for calculating the conditional likelihood covariance, and the objective for each component. This could perhaps be further improved by visualizing the kernel, Fourier transform, covariance matrix, etc. as tensors in more of a proper network diagram. The empirical results are generally quite clear, although I would consider several changes. Table 1 should be converted into plots, or at least the best results should be bolded. Similarly, the results in Table 2 could be bolded. It may also be helpful to visualize the results in frequency space, showing the spectrum of the datasets and comparing with that of the reconstructed and generated results. One would expect to see that the proposed result better matches the power of the higher frequencies as compared with baselines.

**Quality** - The quality of the paper is high. The authors take concepts from the image processing literature and show how to incorporate these into VAEs in a principled way. The mathematics are clear throughout. The end result is a valid probabilistic model with added de-blurring capabilities. The authors demonstrate the benefits of this approach on relevant datasets, comparing against multiple relevant baselines, while holding other factors (i.e., model architecture) constant. Results are reported using a variety of metrics, where the proposed method compares favorably. Thus, this paper takes a well-formulated idea and demonstrates that it improves performance. The authors may consider adding additional results analyzing precisely how the proposed method improves performance. Likewise, results on additional datasets would expand the appeal of the paper beyond generating faces and MRI images.

**Novelty** - The paper is somewhat novel. Many previous works have considered ways to better capture the dependencies between input dimensions in VAEs. Some of these works fall into other model classes, like including autoregressive models at the output, while others offer up heuristic approaches, e.g., perceptual losses. This paper takes the idea of de-blurring/sharpening from the image processing literature and demonstrates how to incorporate these into VAEs. The mathematical ideas, though not overly complex, are not generally found within the VAE literature. Further, there are a significant number of choices to be made in properly formulating the approach, i.e., this isn’t simply a matter of tweaking some previous proposed model/method.

**Reproducibility** - I suspect that the results are reproducible, as the authors present comparisons with multiple baselines on multiple datasets. And the results of the proposed method do, qualitatively, look better than the baselines. However, the authors may wish to improve the analysis of their results by running multiple seeds and reporting confidence intervals. Likewise, especially with modular techniques like this, it may help to use a standard, previously-proposed architecture when running experiments, allowing for readers to more easily compare results across papers. Finally, one of the benefits of the proposed approach is that one can still retain the proper probabilistic model formulation, so it may be useful to also report log-likelihood results.

**Strength And Weaknesses:**

**Strengths**

- **Principled, well-formulated approach that connects two sub-fields to tackle a known problem.** The authors rigorously demonstrate how to incorporate learned, de-blurring convolutional kernels into the conditional likelihood of VAEs, such that the probability density is still valid. While various previous works have proposed heuristics for improving the blurring in VAEs, even looking at spatial frequencies, to the best of my knowledge, none of these previous works have incorporated these aspects in a rigorous probabilistic modeling framework. The end result is a clear demonstration of how to connect ideas from low-level image processing to VAEs, which then be built upon for further in future work. Further, this approach attempts to directly address a known issue with VAEs (blurring) using a principled approach: adding spatial dependencies to the conditional likelihood.

- **Fairly clear presentation.** For the most part, the authors do a great job of explaining the motivation and mechanics behind their approach. They start by discussing why it is that high-frequency components tend to be neglected in VAEs and other maximum-likelihood generative models. They then present basic concepts from image processing, i.e., de-blurring. They then walk through how to incorporate this operation into the covariance of the conditional likelihood in VAEs, covering the key components and hyperparameters. I especially appreciated the diagram outlining the approach, which was quite clear. While I still have several questions about the formulation, the paper’s presentation is in good shape overall.

- **Reasonable empirical evaluation.** The empirical evaluation appears to be sufficient to demonstrate meaningful benefits over previous similar methods on several representative datasets. The authors evaluate standard metrics to quantify their performance improvements. The provided ablation evaluation is also helpful to assess the dependence on various hyperparameters.


**Weaknesses**

- **Could be scaled up further.** The authors demonstrate their method in the setting of single-level convolutional —> fully-connected —> convolutional VAEs. Most competitive VAEs utilize multiple levels of latent variables, e.g., VDVAE (Child, 2021). In principle, this method could be combined with such higher capacity models to possibly provide even further improvement. It would be useful to assess whether this de-blurring technique is still useful in this higher capacity setting. Likewise, in the natural image setting, the authors only demonstrate results on CelebA (although they use two versions of the dataset). I would have expected results on other benchmark datasets, particularly CIFAR10. Generating faces using datasets that have fairly normalized poses feels like perhaps too specific of a setting to make convincing arguments about modeling sharper edges in natural images.

- **Some aspects a bit unclear.** While the method itself appears to be well-formulated, some aspects were a bit unclear to me. For instance, why do we need a second objective to learn the convolutional filter? Shouldn’t this already be rolled into the maximum likelihood objective? Is this primarily for optimization purposes? What dictates the form of this objective — the form of the conditional likelihood? Could we have multiple stages of de-blurring? How does this compare with, for example, using an autoregressive or flow-based conditional likelihood? These alternative forms of densities are also capable of capturing low-level pixel dependencies while still maintaining a valid log-likelihood.

- **Seems somewhat specific to Gaussian conditional likelihoods.** The authors present their approach purely using Gaussian conditional likelihoods. While this is typically a reasonable assumption, this also seems to limit the scope of the approach. For instance, many higher-capacity VAEs utilize discretized mixture of logistic distributions (VDVAE; Child, 2021). While I think it may be possible to apply this technique in these other settings, this is left for future work.

**Summary Of The Paper:**

The authors propose a method for reducing blur in VAEs by explicitly incorporating a de-blurring convolutional operation, which enters the covariance of the conditional likelihood. This allows the method to upweight errors at higher frequencies. In the proposed approach, the conditional likelihood is evaluated in frequency-space, with the covariance determinant calculated analytically. The de-blurring kernel is calculated per-image as a function of the latent variables. Notably, the entire approach still optimizes the ELBO, unlike other heuristic methods. Experiments are performed on CelebA (high resolution and low resolution versions) and a medical image dataset. The proposed method outperforms similar approaches in terms of various metrics, although the reconstructions and generations still seem somewhat blurry.

**Summary Of The Review:**

The paper takes a principled idea, de-blurring, and shows how to incorporate this into VAEs while retaining a proper probabilistic model. The proposed method is evaluated on multiple datasets, comparing against relevant baselines. For these reasons, I feel that this paper warrants publication.

---

> ### Author Response · Authors · 2022-11-15
> **Response to Reviewer q8AD**
>
> We thank the reviewer for the very thorough review and appreciate the suggestions. We are glad to see that the reviewer thinks that the approach is principled and well-formulated in addition to showing reasonable empirical validation. Here we will address some issues which the reviewer pointed out.
>
> **Response 1: Most competitive VAEs utilize multiple levels of latent variables**
>
> The final output of powerful hierarchical VAE models indeed show impressive sharp results. This however often comes at a higher computational cost. Our focus here was to improve the more basic VAE model that requires lower computational cost and has less parameters. Nevertheless, we believe that our loss could potentially also have some interesting benefits for more powerful VAE methods. In hierarchical models it has been observed that some latent units can become inactive [Sonderby et al. 2016] during training. In addition, even though the final output image is sharp with many details, often the latent variables at lower resolution, show an absence of details as shown in Figure 4 in [Child et al. 2021]. Whereas our method is focused on penalizing blur, missing details also lead to a large blur factor in our optimization for k (equation 10). Thus it could be interesting to investigate whether incorporating our proposed optimization for latent variables at lower resolution could help enforcing these variables to encode more details. We intend to explore this further in the future.
>
> **Response 2: Results on CIFAR10**
>
> We gladly added results for CIFAR10 to the appendix B of the article. In Table 5 we compared our method with the L2, focal frequency and cross-entropy reconstruction loss and show that for the reconstruction metrics PSNR, SSIM, LPIPS and FID_recon we outperform other methods. Especially for the LPIPS metric (lower is better) we achieve a score of $0.12\pm4.9e^{-2}$ compared to $0.21\pm6.6e^{-2}$ for the second best performing method (L2). Thus, our approach extends beyond celebA edges to other natural edges. This is also visible in the qualitative results provided for CIFAR10 in Figure 5, where our method is able to reconstruct more details in the images. The only metric where our proposed approach performs worse is on the FID score for generated samples. However, the higher FID score for generative samples of FFL and CE are not reflected qualitatively in Figure 5, where all methods struggle to produce good samples.
>
> **Response 3: Why do we need a second objective to learn the convolutional filter? Shouldn’t this already be rolled into the maximum likelihood objective? Is this primarily for optimization purposes? What dictates the form of this objective — the form of the conditional likelihood?**
>
> We can gladly elaborate our choice of probability model. Our probability model for the decoder is in terms of $x$ and $z$ and the parameters $\theta$ (equation 11) since we assume $G_{\gamma}$ to be fixed during this optimization step. The final probability model therefore is obtained when $k$ is finally fixed, and the ELBO corresponding to that $k$ is optimized. Whereas in principle it would be possible to include $k$ as an additional variable in this optimization step, and therefore as a part of another probabilistic model, we wanted to guide the ELBO optimization to a solution favoring sharp images. Thus in this work we opted for a different optimization objective in determining $k$ that is explicitly designed to reduce blur and effectively trying to highlight the areas that do not match using the L2 optimization. This guidance of the ELBO optimization through $k$ with a separate optimization objective thus does not allow one concise probability model for the decoder including all variables, but was designed in this way on purpose.
>
> **Response 4: The authors present their approach purely using Gaussian conditional likelihoods**
>
> The Gaussian conditional likelihood indeed provided a very neat integration through the covariance matrix. In this work we exploited this insight. For classical VAEs our approach seems to be powerful enough to outperform other probability models. It would be interesting to see how our loss performs in hierarchical models and if a discretized mixture of logistic distributions are superior in that scenario and is left for future work.
>
> **Response 5: Could we have multiple stages of de-blurring?**
>
> If we understand correctly, the question from the reviewer is whether we can also imagine not having alternate optimization between equations 11 (ELBO) and 12 (Finding $k$) but only optimizing for $k$ after every 1000 ELBO iterations for example. This is definitely also a possibility. Alternate optimization is not required and one can imagine having multiple stages where $k$ is optimized and then to optimize the ELBO again with respect to the new $k$. This would be a special case of our approach.

---

> > ### Author Response · Authors · 2022-11-15
> > **Response to Reviewer q8AD**
> >
> > **Response 6: How does this compare with, for example, using an autoregressive or flow-based conditional likelihood?**
> >
> > Powerful autoregressive and flow models such as PixelCNN++[Salimans et al., 2017] and  Flow++ [Ho et al., 2019] achieve negative log likelihoods similar to that of state of the art hierarchical VAEs. We therefore assume that they will achieve better reconstruction and sample generation results than our approach integrated into a classical VAE. We did not compare to them, since our method is specific to VAEs that use low dimensional latent spaces and we focused on evaluating the benefit of our approach. If our proposed approach could benefit hierarchical VAEs (Response 1) this would be interesting to compare the potential of our approach among a broader set of generative models.
> >
> > **Response 7: Improving the diagram of the model/method visualizing the kernel, Fourier transform, covariance matrix, etc. as tensors.**
> >
> > We thank the reviewer for the suggestion. We created a new diagram where we tried to visualize some key elements of the diagram. We replaced the previous diagram with the new one and it should be visible in the current uploaded article version in Figure 1.
> >
> > **Response 8: Table 1 should be converted into plots, or at least the best results should be bolded. Similarly, the results in Table 2 could be bolded.**
> >
> > We thank the reviewer for pointing out the possible improvements for Tables 2 and 3 with our quantitative results. We added the standard deviation and bolded the best results for Tables 2 and 3 in the main paper and for all the Tables in the appendix. This should be visible in the current version of the paper.
> >
> > **Response 9: It may also be helpful to visualize the results in frequency space, showing the spectrum of the datasets and comparing with that of the reconstructed and generated results.**
> >
> > We agree with the reviewer that this would indeed be an interesting comparison. We gladly added some visual results to the appendix B in Figure 6. We show the absolute value of the Fourier transform in log scale for the reconstructions and samples in the top and bottom row, respectively. To obtain the shown images we average over 128 individual reconstructions of the testset and generated samples of the CelebA64 dataset, respectively. To the right of the figure the corresponding image for the data distribution is shown. In addition, we report the mean difference between each shown image with respect to the image created from the true data distribution. Both qualitatively and quantitatively our approach can better replicate the frequency space of the data distribution compared to the baselines L1, L2 and FFL.
> >
> > **Response 10: The authors may wish to improve the analysis of their results by running multiple seeds and reporting confidence intervals.**
> >
> > Following this suggestion, we ran our method with 5 different seeds for the CelebA64 dataset. We report a standard deviation of 0.018, 0.00089, 0.00067, 0.00057 and 0.0014 for the metrics PSNR, SSIM, LPIPS, FID_recont and FID_gen, respectively. Given these low numbers we only used one run for our results. To enhance reproducibility, we will also make our code available. In addition, we also added results for CelebA64 with a different base architecture in the appendix A, Table 4, where it can be seen that also with this new architecture, which has less parameters, results favor our proposed approach.
> >
> > **Response 11: It may be useful to also report log-likelihood results.**
> >
> > As our proposed method mainly focuses on improving the visual results of reconstructions and generations, we focused on visual metrics to show our improvement. For completeness we added the ELBO results of our method and L2 in the appendix B in Table 6 for the CelebA64 dataset.

---

> > > ### Comment · Reviewer_q8AD · 2022-12-05
> > > **Response to authors**
> > >
> > > Thank you for your detailed reply, and thank you for addressing several of the suggestions that I raised regarding the paper. I feel that the paper warrants publication -- however, as my score was already fairly high (an 8), I intend to keep it the same as before.

---

### Official Review · Reviewer_QwSD · 2022-10-24

**Confidence:** 2
**Correctness:** 3
**Technical Novelty And Significance:** 3
**Empirical Novelty And Significance:** 2
**Recommendation:** 6

**Clarity, Quality, Novelty And Reproducibility:**

The paper is well written and the proposed reconstruction term seems novel. The results in the table don't have variance, thus their significance is hard to determine.

**Strength And Weaknesses:**

Strength:
1. The paper provide a detailed ablation study on the choice of Sigma and C.
2. The proposed method seems quite novel and interesting.

Weakness:
1. The paper only compares with other algorithms that have alternative reconstruction term, but did not provide comparison with other SOTA VAE variations. A simple example is the beta VAE which weighs the KL divergence. In general, we could either weighs the reconstruction term or the KL term of the VAE. I think for the very least, comparison with other KL divergence weighting algorithm would be meaningful.
2. in terms of experiment results, it looks like for CELEBA 256 by 256 dataset, the proposed reconstruction term doesn't seem to to be better than the cross entropy reconstruction in a few metrics. Although for the medical dataset, there seems to be improvements. The significance of the improvements are hard to determine because no standard deviations are provided. The reconstruction images of the medical dataset from L2 and proposed method seem quite similar visually. The overall the benefits of the algorithm are not very convincing.


**Summary Of The Paper:**

The paper proposed a new reconstruction term that minimize the blurriness of the VAE while still maintaining the mathematical connection to the original ELBO objective. The paper provides an ablation study of several hyperparameters and tested on dataset such as CELEBA and HCP medical dataset while comparing to some of the other recently proposed reconstruction term.

**Summary Of The Review:**

I think the paper is interesting and the proposed algorithm is novel but motivation for the new reconstruction term is weak because its improvement is hard to determine.

---

> ### Author Response · Authors · 2022-11-15
> **Response to Reviewer QwSD**
>
> We thank the author for their time and suggestions. We are pleased to see that our work is seen as novel and interesting and that the ablation study was conducted well. Here we will address some issues which the reviewer pointed out.
>
> **Response 1: Comparison to Beta-VAE**
>
> We agree with the reviewer that the weighting of the KL-term has an influence on the performance of the VAE. To account for this factor we evaluated the methods with different KL weighting terms on a validation set of 2.5k images and reported the results where the FID score for generative samples was minimized to ensure high visual quality. We added a clarification of this point to the main paper in section 4.4.
>
> **Response 2: Cross entropy reconstruction better in a couple of metrics for CelebA256**
>
> It is true that only for the CelebA256 dataset in terms of the PSNR and SSIM metrics for the reconstructions of the VAE we perform slightly worse compared to cross-entropy. However on the same dataset we perform better than cross-entropy for visual perception metrics (LPIPS, FID), which sharpness is a big part of, in both reconstruction and generation. Furthermore, considering all metrics and all datasets we outperform cross-entropy as a reconstruction loss.
>
> **Response 3: The significance of the improvements are hard to determine because no standard deviations are provided**
>
> We agree with the reviewer and added standard deviations for the metrics in Tables 2 and 3 in the main paper as well as for the new results added in the appendix. We note that for the FID we only obtain a single score, since we evaluate between two sets of images, and thus report no standard deviation. In addition, we bolded the result of the best performing method for each specific metric.
>
> **Response 4: The reconstruction images of the medical dataset from L2 and proposed method seem quite similar visually**
>
> We thank the reviewer for pointing this out. We investigated this and realized that we mistakenly also used L2 images for our method in Figure 4. We apologize for this and the correct version should now be visible in the newly uploaded paper. The difference between our method and L2 is also clear now, as we can observe that our method outputs sharper edges around the brain lobes enabling more detail on this challenging dataset.
>
> **Response 5: Motivation for the new reconstruction term is weak because its improvement is hard to determine**
>
> In addition, to the above mentioned improvements we added several new results to the appendix to further show our improvement over other baselines. In appendix A, Table 4, we report the performance of our method compared to several baselines on the CelebA64 dataset, but using a different architecture. It can be seen that also with this new architecture, which has less parameters, results favor our proposed approach. Furthermore, we provide quantitative and qualitative results on the CIFAR10 dataset in Table 5 and Figure 5, respectively, in appendix B. In Table 5 we show that for the reconstruction metrics PSNR, SSIM, LPIPS and $FID_{recon}$ we outperform other methods. Especially for the LPIPS metric (lower is better) we achieve a score of $0.12\pm4.9e^{-2}$ compared to $0.21\pm6.6e^{-2}$ for the second best performing method (L2). Thus, our approach extends beyond celebA images to other natural images. This is also visible in the qualitative results provided for CIFAR10 in Figure 5, where our method is able to reconstruct more details in the images. The only metric where our proposed approach performs worse is on the FID score for generated samples. However, the higher FID score for generative samples of FFL and CE are not reflected qualitatively in Figure 5, where all methods struggle to produce good samples. Lastly, by minimizing the blur of the VAE we aim to better match the frequency spectrum of reconstructions and samples to that of the data. In appendix B, Figure 6, we show the absolute value of the Fourier transform in log scale for the reconstructions and samples in the top and bottom row, respectively.  To obtain the shown images we average over 128 individual reconstructions of the testset and generated samples of the CelebA64 dataset, respectively. To the right of the figure the corresponding image for the data distribution is shown. In addition, we report the mean difference between each shown image with respect to the image created from the true data distribution. Both qualitatively and quantitatively our approach can better replicate the frequency space of the data distribution compared to the baselines L1, L2 and FFL.

---

### Official Review · Reviewer_Hcu2 · 2022-10-24

**Confidence:** 4
**Correctness:** 4
**Technical Novelty And Significance:** 3
**Empirical Novelty And Significance:** 3
**Recommendation:** 6

**Clarity, Quality, Novelty And Reproducibility:**

The paper is well organised and clearly written. I believe that the missing architecture details of the the decoder/encoder pairs do not hinder its reproducibility. However, I would have appreciated to see a short explanation for the gradient computation of the Fourier domain terms in (12).

**Strength And Weaknesses:**

Paper strengths:
The authors propose a weighted deblur with more focus on high frequency components in the Fourier space along with a decoder that represents a multivariate Gaussian with non-identity covariance matrix obtained from the blur kernel. This includes a reasonable approximation for computing the determinant of the latter. It is interesting to see that this leads to an VAE objective with parts of it defined in the Fourier space.

The experiments are well designed and convincing. They clearly show that the proposed VAE extension
successfully decreases the blur of the reconstructed/generated images. The authors provide an ablation study for the model hyper-parameters. A comparison with other existing methods shows competitiveness of the proposed approach.

Paper weaknesses:
The main weakness in my opinion is that the authors give no concise probability model for the decoder in terms of $x$, $\hat{x}$ and $z$, parametrised by  $\theta$ and the blur kernel $k$. This is perhaps the reason why the learning of the model splits into two optimisation tasks with inter-connected objectives (11) and (12).

The practical value of the approach is in my view somewhat restricted because it is well known that more complex VAE models, like hierarchical VAEs or VAEs with normalising flows usually do not show this blur degradation in the reconstructed/generated images. On the other hand,it is known (see [1,2]) that ELBO learning for standard VAEs introduces a bias towards consistent encoder/decoder pairs which correspond to EF-harmoniums as joint models. It would be therefore interesting, to explore whether the "explaining away" of high frequency image components could be attributed to the restricted expressive power of EF-harmoniums.

[1] Welling et al, NeurIPS 2005

[2] Shekhovtsov et al., ICLR 2022

**Summary Of The Paper:**

The composition of the encoder and decoder mapping of standard Gaussian variational autoencoders (VAE) is known to "explain away" high frequency image components as independent Gaussian noise. The authors propose to address the resulting blur in reconstructed and generated images directly, by adding  a blur kernel to the model parameters, which depends on the latent code. They propose a coordinate block descent approach for solving the optimisation w.r.t. the standard VAE parameters and the parameters of an additional network that maps latent codes to blur kernels. Experiments show that this proposed extension of Gaussian VAEs successfully decreases the blur of the reconstructed/generated images.

**Summary Of The Review:**

The paper proposes an interesting VAE extension that directly penalises the generation of blurry images and shows experimentally that it decreases the blur of the reconstructed/generated images. The main weakness of the manuscript is in my view a missing concise probability model for the extended VAE, which is perhaps the reason why the learning task is formulated in terms of two optimisation tasks with interdependent objectives and solved approximately by coordinate block descent.

---

> ### Author Response · Authors · 2022-11-15
> **Response to Reviewer Hcu2**
>
> We thank the reviewer for their time and suggestions for the paper. We are glad to see that they thought it was an interesting idea and that the experiments were well designed and convincing. Here we will address some issues which the reviewer pointed out.
>
> **Response 1: No concise probability model for the decoder in terms of $x$, $\hat{x}$ and $z$, parametrized by $\theta$ and $k$**:
>
> We can gladly elaborate our choice of probability model. Our probability model for the decoder is in terms of $x$ and $z$ and the parameters $\theta$ (equation 11) since we assume $G_{\gamma}$ to be fixed during this optimization step. The final probability model therefore is obtained when $k$ is finally fixed, and the ELBO corresponding to that $k$ is optimized. Whereas in principle it would be possible to include $k$ as an additional variable in this optimization step, and therefore as a part of another probabilistic model, we wanted to guide the ELBO optimization to a solution favoring sharp images. Thus in this work we opted for a different optimization objective in determining $k$ that is explicitly designed to reduce blur and effectively trying to highlight the areas that do not match using the L2 optimization. This guidance of the ELBO optimization through $k$ with a separate optimization objective thus does not allow one concise probability model for the decoder including all variables, but was designed in this way on purpose.
>
> **Response 2: Limited practical value since more powerful VAE models do not show this blurriness**
>
> The final output of powerful hierarchical VAE models indeed show impressive sharp results. This however often comes at a higher computational cost. Our focus here was to improve the more basic VAE model that requires lower computational cost and has less parameters. Nevertheless, we believe that our loss could potentially also have some interesting benefits for more powerful VAE methods. In hierarchical models it has been observed that some latent units can become inactive [Sonderby et al. 2016] during training. In addition, even though the final output image is sharp with many details, often the latent variables at lower resolution, show an absence of details as shown in Figure 4 in [Child et al. 2021]. Whereas our method is focused on penalizing blur, missing details also lead to a large blur factor in our optimization for $k$ (equation 10). Thus it could be interesting to investigate whether incorporating our proposed optimization for latent variables at lower resolution could help enforcing these variables to encode more details. We intend to explore this further in the future.
>
> **Response 3: Could the "explaining away" of high frequency image components be attributed to the restricted expressive power of EF-harmoniums?**
>
> We thank the reviewer for pointing out the very interesting work by [Shekhovtsov et al. 2022]. In their work the authors point out that using a Gaussian probability model for the decoder with a fixed variance has a tight optimal VAE where the joint distribution is multivariate Gaussian. As the reviewer also points out, standard VAEs have a bias towards tight optimal VAE solutions. A multivariate Gaussian as joint distribution is however restrictive and would struggle capturing the complexity of realistic images. The complexity of realistic images are arguably further increased by high frequency details such as sharp edges. Thus we tend to agree with the reviewer that this limit in expressive power leads to not being able to capture high frequency details. Shekhovtsov et al. further points out that the expressive power of the tight optimal VAE can be greatly increased by allowing the variance of the decoder probability model to be dependent on z. This reveals an interesting link to our work. In our approach the co-variance matrix is dependent on z as shown in equation 11. This dependency thus enables more expressive joint distributions for the tight optimal VAE and could be another reason why our method is better able to capture high-frequency details in images.
>
> **Response 4: Missing architecture details of the decoder/encoder pairs**
>
> We apologize for the missing architecture details. We added the details to the appendix A. In addition, we added results in the appendix A, Table 4, showing the performance of our method compared to other baselines using another model architecture, which is also described in detail in the appendix A. It can be seen that also with this new architecture, which has less parameters, results favor our proposed approach.
>
> **Response 5: Short explanation for the gradient computation of the Fourier domain terms in (12)?**
>
> In equation 12 we are not using Fourier domain terms, we assume the reviewer was referring to equation 11 above. For the Fourier transform we used the two dimensional Fast Fourier transform as implemented in the PyTorch library (torch.fft.fft2). To compute the gradients we used PyTorch’s automatic back-propagation algorithm.

---

> > ### Comment · Reviewer_Hcu2 · 2022-11-18
> > **Response**
> >
> > Thank you for your detailed response to the raised questions and issues. I tend to keep my recommendation because I still believe that the proposed method can be improved by a concise decoder model that involves the blur kernel as a parameter and leads to a single objective function.

---

> > > ### Author Response · Authors · 2022-11-24
> > > **Response to Reviewer Hcu2**
> > >
> > > We thank the reviewer for taking our response into consideration. To investigate the reviewer’s remaining concern, we ran experiments for a naive concise decoder model on the CelebA64 dataset. More specifically, we optimized the parameters of the kernel generator, $G_{\gamma}$, within the ELBO objective (equation 11 in the paper) instead of using a separate objective (equation 12). We observed that this formulation does improve performance with respect to baselines such as L2, which can be motivated by the increased complexity of the tight optimal VAE as further explained in the previous response 3. This concise formulation also performed similarly for the reconstruction metrics PSNR and SSIM with values of 23.15 and 0.7345, respectively, compared to our approach. However, with respect to perceptual metrics such as LPIPS (lower is better) and FID (lower is better) our approach (left) outperformed with a margin the concise decoder model (right) presented above for both reconstructions and generations.
> > >
> > > LPIPS: 0.1254$\pm$5.7$e^{-2}$ vs. 0.1508$\pm$6.2$e^{-2}$ |   FID_recon: 0.0364 vs. 0.0481  |  FID_gen: 0.0536 vs. 0.0683
> > >
> > > These results support our argumentation in the previous comment (Response 1) that the second objective is a useful guidance for the ELBO to obtain better perceptual results. This guidance is through focusing on the frequency components that lead to an error and can be compensated with a blurring kernel. We cannot say that this behavior cannot be achieved using a joint model. Indeed such a model would be very interesting. However, it is surely not trivial to construct such a model.

---

### Comment · Area_Chair_YeF3 · 2022-11-15
**Please respond to rebuttal**

Dear reviewers,

Your response to the authors' rebuttal would be highly appreciated.

Kind regards,
Your AC

---

### Decision · Program_Chairs · 2023-01-20

**Decision:**

Accept: poster

**Justification For Why Not Higher Score:**

There were some complaints about the lack of concise presentation of the probabilistic model, that the method seems somewhat specific to Gaussian likelihoods, and that the model on which the method is tested is small by contemporary standards.

**Justification For Why Not Lower Score:**

This seems like a decent paper. The reviewers commend the principled approach, and the experiments against reasonable baselines.

**Metareview: Summary, Strengths And Weaknesses:**

Ratings: 6/6/8
Confidence: 4/2/4

The paper proposes a method for reducing blur in the samples of variational autoencoders (VAEs) by incorporating a de-blurring convolutional operation in the output of the decoder p(x|z), through the covariance of the conditional likelihood. In the approach, the conditional probability p(x|z) is evaluated in frequency space, with the covariance determinant computed analytically. Unlike other heuristic methods, the method still optimizes the ELBO. Experiments show that the method outperforms alternative approaches, although the method was not able to remove all blur.

The reviewers commend the principled approach, and the experiments against reasonable baselines. There were some complaints about the lack of concise presentation of the probabilistic model, that the method seems somewhat specific to Gaussian likelihoods, and that the model on which the method is tested is small by contemporary standards.

**Note From Pc:**

if the above contains the word "oral" or "spotlight" please see: "oral" presentation means -> notable-top-5% and "spotlight" means -> notable-top-25%. As stated in our emails, we are disassociating presentation type from AC recommendations